# Individual Genomic Loci and mRNA Levels of Immune Biomarkers Associated with Pneumonia Susceptibility in Baladi Goats

**DOI:** 10.3390/vetsci10030185

**Published:** 2023-03-01

**Authors:** Ahmed Ateya, Mona Al-Sharif, Mohamed Abdo, Liana Fericean, Bothaina Essa

**Affiliations:** 1Department of Animal Husbandry and Animal Wealth Development, Faculty of Veterinary Medicine, Mansoura University, Mansoura 35516, Egypt; 2Department of Biology, College of Science, University of Jeddah, Jeddah 21589, Saudi Arabia; 3Department of Animal Histology and Anatomy, School of Veterinary Medicine, Badr University in Cairo (BUC), Cairo 11829, Egypt; 4Department of Anatomy and Embryology, Faculty of Veterinary Medicine, University of Sadat City, Sadat City 32897, Egypt; 5Department of Biology and Plant Protection, Faculty of Agricultural Sciences, University of Life Sciences King Michael I, 300645 Timisoara, Romania; 6Department of Animal Husbandry and Animal Wealth Development, Faculty of Veterinary Medicine, Damanhour University, Damanhour 22511, Egypt

**Keywords:** genetic polymorphisms, goat, pneumonia, immune biomarkers, gene expression

## Abstract

**Simple Summary:**

Pneumonia is a commonly encountered problem in small ruminant medicine practice that negatively affects goat breeding. Some of the limitations of the conventional approaches used by animal breeders to promote animal health may be overcome with the use of molecular genetic tools. Using PCR-DNA sequencing in pneumonia-affected and -resistant Baladi goats, SNPs associated with pneumonia resistance/susceptibility were found in the *SLC11A1*, *CD-14*, *CCL2*, *TLR1*, *TLR7*, *TLR8*, *TLR9*, *β defensin*, *SP110*, *SPP1*, *BP1*, *A2M*, *ADORA3*, *CARD15*, *IRF3*, and *SCART1* genes. These indicators’ mRNA levels were different in both the healthy and affected goats. Therefore, to improve the effectiveness of animal selection for innate resistance, finding potential genes linked to pneumonia risk may be crucial.

**Abstract:**

The effectiveness of breeding for inherent disease resistance in animals could be considerably increased by identifying the genes and mutations that cause diversity in disease resistance. One hundred and twenty adult female Baladi goats (sixty pneumonic and sixty apparently healthy) were used in this study. DNA and RNA were extracted from blood samples collected from the jugular vein of each goat. *SLC11A1*, *CD-14*, *CCL2*, *TLR1*, *TLR7*, *TLR8*, *TLR9*, *β defensin*, *SP110*, *SPP1*, *BP1*, *A2M*, *ADORA3*, *CARD15*, *IRF3*, and *SCART1* SNPs that have been previously found to be associated with pneumonia resistance/susceptibility were identified via PCR-DNA sequencing. The pneumonic and healthy goats differed significantly, according to a Chi-square analysis of the discovered SNPs. The mRNA levels of the studied immune markers were noticeably greater in the pneumonic goats than in the healthy ones. The findings could support the significance of the use of immune gene expression profiles and nucleotide variations as biomarkers for the susceptibility/resistance to pneumonia and provide a practical management technique for Baladi goats. These results also suggest a potential strategy for lowering pneumonia in goats by employing genetic markers linked to an animal’s ability to fend off infection in selective breeding.

## 1. Introduction

The domestic goat (*Capra hircus*), one of the original livestock species, is an important component of animal management [1]. The ability to adapt is one of the core qualities of goats; for example, the majority of dairy goat breeds in use today are ancestors of European kinds [2]. There are 4.3 million goats in Egypt. Around the world, goats are raised extensively, especially in poor countries, where they are a valuable resource for the production of goods and a variety of household items [3,4,5,6,7].

Pneumonia is a frequent respiratory disease in the majority of animals worldwide [8]. Acute lung infection, known as pneumonia, restricts oxygen intake due to fluid-filled alveoli, which is a crucial feature of the disease’s pathogenesis [9]. Pneumonia can be caused by a variety of factors, although the most frequent ones are bacterial or viral infections [8]. Traditionally, farm management practices and the creation of vaccines have always been the key priorities for pneumonia prevention. The use of pharmaceutical interventions in the treatment of pneumonia is significantly influenced by the source of the infection; for example, antibiotics are used to treat bacterially generated pneumonia [10]. Even in the industrialized world, where access to routine clinical treatment and antibiotics is typically unrestricted, annual mortality rates from pneumonia are nevertheless significant [11,12]. This calls for a deeper comprehension of the mechanisms behind pneumonia susceptibility and pathogenesis, which might be used to find new cures and guide drug repositioning [13].

Lung infections typically develop as a result of an interplay between pathogenic microorganisms (bacteria, mycoplasmas, viruses, and fungi), host defense, and distressing environmental conditions, such as traffic, crowding, abrupt climate changes, poor ventilation, and malnutrition [14]. The peak seasons for small ruminant pneumonia in Egypt are often autumn and the beginning of winter, spring, and the start of summer [15]. This is a result of the reserve or sudden temperature fluctuations in the weather from warm to cold. Additionally, it has been discovered that there are more dust particles delivering infections deep into the respiratory system in the spring and autumn [16].

A comprehensive strategy for disease control may be found in genetic selection for the improvement of animal health [17]. Due to the complex interactions between pathogens, the environment, and variables specific to animals that favor their development, pneumonia, and other economically significant illnesses of farm animals pose a challenge to their effective management or eradication [18]. Changes in the management, treatment, vaccination, pathogen control, movement control, slaughter, isolation, and quarantine of diseased animals are just a few of the initial methods of control that are no longer effective due to increased pathogen resistance to chemotherapeutic and prophylactic drugs, antibiotic residues in the environment, and animal products, to name a few [17]. The key to the low-cost and effective control of chronic diseases is to make use of host genetic resistance because it does not suffer from these restrictions in the broadest sense [19]. Unfortunately, in order to attain effective control of these disorders, several gene loci must be identified and defined [18]. In order to find disease-resistance markers, quantitative trait loci (QTLs) are mapped and numerous potential gene variations are evaluated for associations with disease resistance [20]. A great deal of attention has been paid to the genetics of innate and acquired immunity related to resistance, and there is now evidence linking changes in some of these genes with resistance to diseases [21].

The diagnosis and management of complex disorders under multigenic control are more challenging than those under unigenic control [22]. Pathogenic and environmental variables also exacerbate the interplay of several host genes [23]. However, this does not rule out the possibility of genetic control; rather, it necessitates a greater understanding of the interrelated factors [24]. Finding host variables that affect the onset and clinical course of pneumonia in small ruminants has been the subject of extensive research [25,26].

The objective of this study was to determine how potential immune markers (*SLC11A1*, *CD-14*, *CCL2*, *TLR1*, *TLR7*, *TLR8*, *TLR9*, *β defensin*, *SP110*, *SPP1*, *BP1*, *A2M*, *ADORA3*, *CARD15*, *IRF3*, and *SCART1*) have an impact on the prevalence of pneumonia resistance/susceptibility in Baladi goats using PCR-DNA sequencing and quantitative real-time PCR.

## 2. Materials and Methods

### 2.1. Ethics Statement

The methods used to collect samples and to care for the animals in this experiment were permitted by the Institutional Committee for the Care and Use of Animals (IACUC) of the Veterinarian Medical Faculty, Damanhour University, Egypt (code DMU/VetMed-2023/001).

### 2.2. Animals and Research Subjects

A total of 120 adult female Baladi goats (60 pneumonic and 60 seemingly healthy), aged 4 years old, weighing 48 ± 3.5 kg, and being raised on a private farm in Damanhour province, Egypt, were employed in this study. The does were housed in shaded, semi-open pens and nourished with 650 g of a concentrate feed mixture (CFM) and 650 g of alfalfa hay per head each day, with unlimited access to water. The CFM was composed of wheat bran (220 kg), soya bean (210 kg), corn (520 kg), sodium chloride (4 kg), calcium carbonate (8 kg), Premix (1.25 kg), Netro-Nill (0.5 kg), and Fylax (0.5 kg). When available, the natural pasture (green vegetation, such as berseem and darawa) was used as feed.

A thorough clinical examination was performed on the goats under investigation in accordance with previously described standard protocols [27,28], and the results were simultaneously recorded. The jugular veins of each doe were punctured to obtain five milliliters of blood. Blood samples were collected from healthy and pneumonic does during active pneumonia. To obtain whole blood for the purpose of extracting DNA and RNA, the samples were placed in vacuum tubes with anticoagulants (sodium fluoride or EDTA).

### 2.3. DNA Isolation and Amplification

Following the established protocol, the extraction of DNA from the blood samples was carried out using a commercial kit, a QI-Aamp DNA Mini Kit (Qiagen, Hilden, Germany) (DNA purification from blood or body fluids). The only samples judged to be appropriate for analyses had DNA levels between 5 and 40 ng/μL and A260/A280 ratios of 1.7 to 1.9, as determined utilizing the DNA quantification and extraction software from Nanodrop (NanoDrop Technologies, Wilmington, DE, USA).

PCR was used to amplify the coding regions of the immune genes (*SLC11A1*, *CD-14*, *CCL2*, *TLR1*, *TLR7*, *TLR8*, *TLR9*, *β defensin*, *SP110*, *SPP1*, *BP1*, *A2M*, *ADORA3*, *CARD15*, *IRF3*, and *SCART1*). *The Capra hircus* sequence available in PubMed was used to design the primer sequences. Appendix A lists the primers that were employed during the amplification.

The mixture for the polymerase chain amplification was passed through a thermal cycler with a concluding capacity of 150 μL. The following ingredients were used in each reaction volume: 75 μL of master mixture (Jena Bioscience, Jena, Germany), 6 μL of DNA, 1.5 μL of each complementary primer, and 66 μL of d.d. water. The reaction mixture spent 4 min at a 95 °C starting denaturation temperature. One-minute denaturation cycles at 95 °C, annealing for one minute at the temperatures shown in Appendix A, extending for one minute at 72 °C, and then extending for a further ten minutes at 72 °C were all included in the 35-cycle procedure. The samples were stored at 4 °C. Following the discovery of representative PCR findings using agarose gel electrophoresis, fragment patterns under UV light were visualized using a gel documentation system.

### 2.4. Sequencing DNA and Detecting Polymorphism

Jena Bioscience # pp-201s/Munich, Hamburg, Germany, provided kits for PCR purification to remove non-specific bands, primer dimmers, and other impurities before DNA sequencing, resulting in the desired PCR product of the expected size [29]. For PCR quantification, using a Nanodrop (UV-Vis spectrophotometer Q5000/Waltham, MA, USA), good quality and good concentrations were obtained [30]. The PCR results bearing the objective PCR product were forward- and reverse-sequenced to search for SNPs in the healthy and pneumonic goats. An ABI 3730XL DNA sequencer was used to sequence these products utilizing the enzymatic chain terminator approach discovered by Sanger et al. [31] (Applied Biosystems, Waltham, MA, USA).

The DNA sequencing data were analyzed using the software programs Chromas 1.45 and BLAST 2.0 [32]. SNPs were found as differences between the immune gene products generated using the PCR and the reference GenBank-sourced gene sequences. The MEGA4 tool uncovered variations in the amino acid sequences of the investigated genes on the basis of a sequence alignment among the does [33].

### 2.5. RNA Isolation and Immune Gene Quantitation

The blood samples from the does under investigation were used to obtain their total RNA via the Trizol reagent, in line with the directions provided by the manufacturer (RNeasy Mini Ki, Catalogue No. 74104). We measured and verified the quantity of the isolated RNA using a NanoDrop^®^ ND-1000 spectrophotometer. The manufacturer’s procedure was used to synthesize the cDNA for each sample (Thermo Fisher, Waltham, MA, USA, Catalog No, EP0441). The expression patterns of the genes *SLC11A1*, *CD-14*, *CCL2*, *TLR1*, *TLR7*, *TLR8*, *TLR9*, *β defensin*, *SP110*, *SPP1*, *BP1*, *A2M*, *ADORA3*, *CARD15*, *IRF3*, and *SCART1* were assessed using quantifiable RT-PCR and the SYBR Green PCR Master Mix (2x SensiFast^TM^ SYBR, Bio-line, CAT No: Bio-98002). With the use of the SYBR Green PCR Master Mix, a comparative amount of mRNA was quantified (Quantitect SYBR green PCR kit, Toronto, ON, Canada, Catalog No, 204141). The *Capra hircus* sequence available in PubMed was used to design the primer sequences (Appendix A). The constitutive control for normalization was the *ß. actin* gene. The reaction mixture contained the following components: 25 µL of total RNA, 0.5 µL of each complementary primer, 8.25 µL of RNase-free water, 0.25 µL of reverse transcriptase, 12.5 µL of Quantitect SYBR green PCR master solution, and 4 µL of Trans Amp buffer. The finished reaction blend underwent a subsequent program in a thermal cycler: reverse transcription at 50 °C for 30 min; preliminary denaturation at 94 °C for 8 min, 40 cycles at 94 °C for 15 s, and the annealing temperatures listed in Appendix A; and elongation for 30 s at 72 °C. In order to confirm the specificity of the PCR product, a melting curve analysis was performed after the amplification stage. The 2^−ΔΔCt^ method was utilized to assess how the expression of each gene varied in the tested sample in comparison to that of the *ß. actin* gene [34,35].

### 2.6. Statistical Analysis

H_0_: The individual genomic loci and gene expressions of *SLC11A1*, *CD-14*, *CCL2*, *TLR1*, *TLR7*, *TLR8*, *TLR9*, *β defensin*, *SP110*, *SPP1*, *BP1*, *A2M*, *ADORA3*, *CARD15*, *IRF3*, and *SCART1* do not account for the resistance/susceptibility to pneumonia in Baladi goats.

H_A_: The individual genomic loci and gene expressions of *SLC11A1*, *CD-14*, *CCL2*, *TLR1*, *TLR7*, *TLR8*, *TLR9*, *β defensin*, *SP110*, *SPP1*, *BP1*, *A2M*, *ADORA3*, *CARD15*, *IRF3*, and *SCART1* account for the resistance/susceptibility to pneumonia in Baladi goats.

The considerable variations in the genes’ SNPs that were found between the investigated does were determined using a chi-square analysis. Utilizing the statistical software application Graphpad, a statistical analysis was carried out for this aim (*p* < 0.05). It was determined whether the difference between the pneumonic and healthy Baladi does was statistically significant using the t-test and version 17 of the Statistical Package for Social Science (SPSS) computer program (SPSS Inc., Chicago, IL, USA). To present the findings, mean and standard error (Mean ± SE) were employed. *p* < 0.05 was used to determine whether the differences were significant.

The relevance of several variables was evaluated using a discriminant analysis model, and the gene expression profile of the examined immune genes was used as an independent variable to categorize the pneumonic and healthy does as a dependent variable. The examined genes’ transcript levels were used to differentiate between the pneumonic and healthy does. The interaction exerted by two components (gene kind and pneumonia incidence) and its influence on transcript levels were examined using a two-way ANOVA and the univariate general linear model (GLM).

## 3. Results

### 3.1. Clinical Findings

Based on health, the examined does were split into two equal-sized groups (60 each). Clinically healthy individuals made up the first group (i.e., those with a normal appetite, a normal body temperature, normal respiratory and pulse rates, bright eyes, and no lacrimal or nasal discharge), which was named the healthy group. Pneumonia was present in the second group (with mucopurulent nasal discharge, hyperthermia, abdominal respiration, weakness, an off appetite, crackles, exercise intolerance, gasping, and wheezing observed via auscultation).

### 3.2. Polymorphisms of Immune Markers

First, the genomic DNA from the investigated does was extracted to amplify immune genes (Appendix A). Purified and good concentration DNA was indicated by Nanodrop (Appendix A). Utilizing the PCR-DNA sequence findings of the *SLC11A1* (523-bp), *CD-14* (538-bp), *CCL2* (534-bp), *TLR1* (471-bp), *TLR7* (398-bp), *TLR8* (799-bp), *TLR9* (460-bp), *β defensin* (253-bp), *SP110* (537-bp), *SPP1* (943-bp), *BP1* (645-bp), *A2M* (325-bp), *ADORA3* (521-bp), *CARD15* (394-bp), *IRF3* (468-bp), and *SCART1* (475-bp) genes, the DNA base sequence SNPs associated with pneumonia were found to vary between the healthy and affected Baladi does. There were no common SNPs between healthy and pneumonic does. The nucleotide sequence variations between the immune genes examined in the does in this study and the reference GenBank-sourced gene sequences were applied to approve all the discovered SNPs (Appendix A). Based on the chi-square analysis of the discovered SNPs, the healthy and pneumonia-affected does showed substantially different frequencies of the studied genes (*p* ˂ 0.05) (Table 1). The immunological indicators under examination all contained the changes described in Table 1 in the exonic region, resulting in coding mutations between the pneumonic does and the healthy does.

### 3.3. Immune Markers’ Patterns of Gene Expressions

The evaluated indicators’ gene expression profiles are shown in Figure 1. The expression levels of *SLC11A1*, *CD-14*, *CCL2*, *TLR1*, *TLR7*, *TLR8*, *TLR9*, *β defensin*, *SP110*, *SPP1*, *BP1*, *A2M*, *ADORA3*, *CARD15*, *IRF3*, and *SCART1* were considerably greater in the diarrheic Baladi goats than in the healthy ones.

The mRNA levels of the investigated indicators were strongly influenced by the type of gene and pneumonia resistance/susceptibility. For every gene analyzed in the pneumonic does, *IRF3* exhibited the greatest level of mRNA (2.51 ± 0.11); *ADORA3* had the lowest level (1.62 ± 0.16). The gene *CD14* exhibited the highest possible quantity of mRNA amongst all the genes inspected in the healthy does (0.59 ± 0.08), whereas *ADORA3* had the lowest (0.38 ± 0.07).

## 4. Discussion

### 4.1. Individual Genomic Loci of Immune Markers Associated with Pneumonia Susceptibility

It is obvious that we still do not fully understand how genetics and epigenetics affect pneumonia in farm animals [36]. Using traditional breeding techniques of selecting disease features with high heritability has been beneficial in reducing the prevalence of various infectious diseases in livestock [37]. The majority of disease features, however, have relatively low heritability, making it difficult to achieve genetic control through breeding [37]. There is optimism that using molecular data will truly help to significantly reduce the number of animal diseases with the advancement of molecular genetic technology and our knowledge of farm animal genomes [38]. The availability of genome sequences and the high-resolution molecular mapping of specific livestock species has allowed for new insights into the genetic causes of infectious diseases [39]. Infection control has been made possible via the identification of the genes and causal SNPs responsible for particular disease traits, with the aid of the location of significant QTLs for those traits [40].

Finding immunological genomic regions that have a quantitative impact on a trait is critical for understanding the genetic architecture of a feature linked to pneumonia susceptibility. In this study, *SLC11A1*, *CD-14*, *CCL2*, *TLR1*, *TLR7*, *TLR8*, *TLR9*, *β defensin*, *SP110*, *SPP1*, *BP1*, *A2M*, *ADORA3*, *CARD15*, *IRF3*, and *SCART1* were molecularly characterized in pneumonia-free and pneumonia-affected Baladi goats. The results demonstrate differences in the SNPs between the two groups. The SNP distribution among the investigated does was significant (*p* ˂ 0.05) according to the chi-square test. It is imperative to emphasize that the polymorphisms discovered and available herein provide additional data about the examined genes when contrasted with those in the associated GenBank-sourced sequences.

Immune-function-related gene mutations have been connected to the susceptibility of animals to pathogenic microorganisms [41,42]. We think that this is the first investigation to find nucleotide sequence variants in immune system genes (*SLC11A1*, *CD-14*, *CCL2*, *TLR1*, *TLR7*, *TLR8*, *TLR9*, *β defensin*, *SP110*, *SPP1*, *BP1*, *A2M*, *ADORA3*, *CARD15*, *IRF3*, and *SCART1*) as potential candidates for pneumonia resistance/susceptibility in Baladi goats. However, the candidate gene method was utilized to evaluate ruminant susceptibility to pneumonia. For example, in sheep, a link between *MHC-DRB1* genotypes for mycoplasma ovipneumonia resistance or susceptibility has been documented [25,43]. Additionally, it has been established that progressive pneumonia virus susceptibility is affected by the effects of ovine *TMEM154* gene polymorphisms after normal exposure [26]. In humans, *TLRs* gene polymorphisms have been reported to be associated with pneumonia [44].

### 4.2. Transcript Levels of Immune Markers Associated with Pneumonia Susceptibility

As a heritable endophenotype, transcript abundance is influenced by chromosomal polymorphisms, according to the theory of hereditary genomics [45]. The latter approach supports the notion that integrating information on the mRNA levels of investigated genes and chromosomal variations may improve our comprehension of the genetics underlying the development of disease [46]. Quantitative trait loci (QTLs) for expression are polymorphisms connected to mRNA quantification [47]. The present work postulated that an individual’s hereditary phenotypic variation in the response of mRNA levels to pneumonia incidence may have an impression on the disease course. Real-time PCR was conducted to quantify the expression levels of immunological markers (*SLC11A1*, *CD-14*, *CCL2*, *TLR1*, *TLR7*, *TLR8*, *TLR9*, *β defensin*, *SP110*, *SPP1*, *BP1*, *A2M*, *ADORA3*, *CARD15*, *IRF3*, and *SCART1*) in Baladi goats with and without pneumonia resistance. Our research revealed that the resistant Baladi goats expressed lower levels of immune genes than the pneumonic goats. As a means of remedying the shortcomings of past research, gene expression and genetic SNP markers were used to assess gene polymorphisms. Therefore, the processes that were studied to control the immunological indicators in both the healthy and pneumonic does are well recognized. To the best of our knowledge, this work is the first to thoroughly analyze the gene expressions of the immunological markers associated with the risk of goat pneumonia. However, gene expression profiling revealed that Toll-like receptors (TLRs) and complement genes were associated with infectious pneumonia in sheep [48].

SLC11A1, a trans-membrane protein, is one of the best-known probable potential genes for innate immunity against a number of intracellular infections [49]. An essential component of innate immunity is the cluster of differentiation 14 (CD14). One of the most crucial molecules that binds and neutralizes bacterial endotoxins is the anti-bacterial peptide CD14 [50]. Lipopolysaccharide (LPS) and peptidoglycan, two of the most prevalent components in the bacterial cell wall, both function as receptors for CD14 [51]. Leucocytes are thought to be transported to the mammary glands by chemokines and their receptors, and they may significantly affect how the host immune system responds to both acute and chronic intramammary infections [52]. CCL2 works as a potent chemokine for monocytes and neutrophils by ligating with the CCR2 receptors expressed on their surface [53].

Pattern-recognition receptors (PAMPs) are distinguished from other molecules by their leucine-rich repeat (LRR) domains, which are essential structural components and are reported to make up the majority of the ectodomains of TLR molecules [54]. In a previous study, when non-synonymous SNP prevalence was evaluated among TLR coding sequences from different animals, it was discovered that the sequences encoding LRR domains are more abundant in non-synonymous SNPs [55]. A molecular system’s ability to detect extracellular infections may be considerably altered by non-synonymous SNPs in LRR domains [56]. Leucocytes and epithelial cells have been thought to be the only sources of β-defensins, which have been thought to be the first line of defense against pathogens [57]. In a previous study, this viewpoint was corroborated by the way in which they responded to various bacterial, viral, and fungal diseases [58]. Goats have two beta-defensins, which have been named goat β-defensin (GBD)-1 and GBD-2. *PreproGBD-1* and *PreproGBD-2* genes are 96.8% and 88.2% identical in terms of nucleotide sequence and amino acid sequence, respectively [59]. Alpha- and beta-defensin genes in mammals have one intron and two exons [60].

The SP110 nuclear body protein (*SP110*) gene is implicated in inducing innate immune responses to combat many intracellular infections [61]. Mycobacterium tuberculosis appears to be under the control of SP110 in host macrophages [62]. According to reports, Mycobacterium avium subspecies paratuberculosis infection is associated with bovine SP110 nuclear body protein (*SP110*) gene polymorphisms [63]. Macrophages and T cells that have been activated generate the cytokine known as secreted phosphoprotein 1 (SPP1), also known as Osteopontin [64]. SPP1 has been identified to be a key player in the T-cell activation process. It boosts T-helper 1 (Th1) cell-mediated immunity via the release of Th1 cytokines and, when T cells are enhanced, attracts macrophages to the infection site [65]. The bactericidal/permeability-increasing endogenous cationic protein (BPI) neutralizes Gram-negative bacteria, endotoxins, and lipopolysaccharides [66]. Inhibiting angiogenesis and the production of inflammatory mediators, boosting complement activation and opsonization for enhanced phagocytosis, and protecting against infection by fungus and protozoan pathogens are biological functions of BPI [66]. As a result, BPI is crucial for the host’s natural defense.

The alpha-macroglobulin (aM) family of proteins, which includes C3, C4, and C5, also includes alpha-2-macroglobulin (A2M) [67]. The innate immune system’s A2M component has been conserved throughout evolution [68]. Additionally, it promotes the growth of macrophages and T cells [69]. The host defense mechanism uses A2M as a non-specific protease inhibitor to inactivate both endogenous and foreign proteases, including metalloproteases and serine, threonine, cysteine, aspartic, and cyclooxygenases [70]. The adenosine A3 receptor, also known as ADORA3, is an adenosine receptor. G-protein-coupled adenosine A3 receptors bind to Gi/Gq and participate in a number of intracellular signaling pathways and physiological processes [71]. They participate in the prevention of neutrophil degranulation in neutrophil-mediated tissue injury, and they exert a sustained cardio-protective role during myocardial ischemia [72].

The CARD15 protein, a pattern-recognition receptor (PRR) with leucine-rich repeats (LRR), a NACHT domain, and two caspase recruitment domains (CARDs), is crucial for both innate and acquired immunity [73]. The interferon regulatory transcription factor (IRF) family includes IRF3 [74]. IRF3 has a profound impact on the innate immune system’s response to microbial infection [75]. A protein that is only present in a specific subset of delta gamma T cells and serves to identify important pathogens is produced by the scavenger receptor family member expressed on T cells (*SCART1*) gene [76]. A genome-wide association analysis revealed that the *SCART1* gene is related to the hoof health trait in Holstein dairy cattle [77].

The development of inflammatory markers driven by lipopolysaccharides or endotoxins may be the cause of the pneumonia-affected does’ noticeably altered immune and antioxidant expression patterns [78]. In macrophages stimulated by LPSs, inflammatory gene expressions triple, mRNA levels rise by about 100 fold, and the release of the protein itself may increase by as much as 10,000 fold [79]. Furthermore, compared to healthy tissues, damaged tissues produce higher free radical responses [80,81]. The susceptible genotypes promote the inflammatory response via the high production of inflammatory markers, whereas the resistant genotypes significantly upregulate the expressions of IL-2 and IL-10 to limit the inflammatory response [82]. Additionally, pneumonia-affected does create a variety of cellular immune components that attach to cell surface receptors and mediate and regulate immune activity and the inflammatory response [83]. According to several studies, there are complex relationships between various cell immunological components. These proteins, for instance, have been found to interact with one another and have an impact on the development of different immune globulins, complements, and acute phase reactive proteins, creating a complex network structure [84,85]. As a result, we anticipate that infectious pneumonia was the primary cause of the majority of the pneumonia cases in our study. Furthermore, the outcomes of our real-time PCR provide a believable indication that animals with pneumonia produce a strong inflammatory response.

## 5. Conclusions

By using PCR-DNA sequencing to analyze immunological markers (*SLC11A1*, *CD-14*, *CCL2*, *TLR1*, *TLR7*, *TLR8*, *TLR9*, *β defensin*, *SP110*, *SPP1*, *BP1*, *A2M*, *ADORA3*, *CARD15*, *IRF3*, and *SCART1*), it was possible to identify an association between pneumonia resistance/susceptibility and nucleotide sequence differences in Baladi goats. Additionally, these immune markers’ mRNA levels varied between the healthy and pneumonic does. With the help of these many functional variants, researchers can better understand how infections work, which could result in the development of novel pharmacological and therapeutic approaches to prevent the spread of diseases. The different transcripts of the immune makers in the pneumonic and healthy does may act as monitors and benchmarks to evaluate health.

## Figures and Tables

**Figure 1 vetsci-10-00185-f001:**
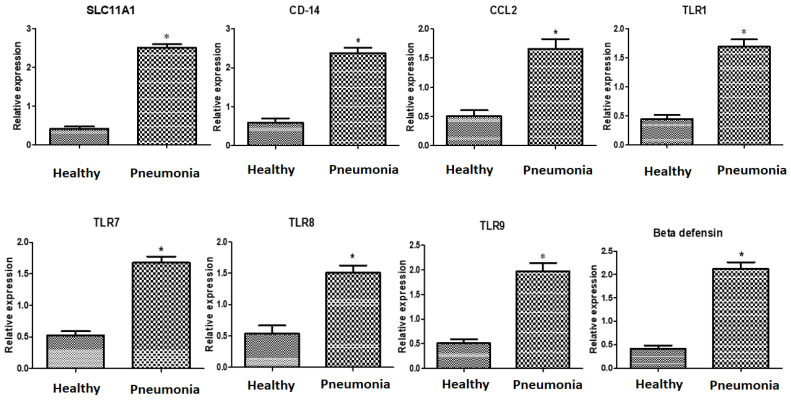
*SLC11A1*, *CD-14*, *CCL2*, *TLR1*, *TLR7*, *TLR8*, *TLR9*, *β defensin*, *SP110*, *SPP1*, *BP1*, *A2M*, *ADORA3*, *CARD15*, *IRF3*, and *SCART1* gene expression comparisons in healthy and pneumonic Baladi goats. * indicates significance when *p* < 0.05.

**Table 1 vetsci-10-00185-t001:** Healthy and pneumonic Baladi goats’ SNP dissemination and type of mutation for the genes in question.

Gene	SNPs	Healthy *n* = 60	Pneumonic *n* = 60	Total *n* = 120	Mutation Type	Number and Kind of Amino Acid	Chi Value	*p* Value
*SLC11A1*	A27T	-	32	32/120	Synonymous	27 S	67.25	<0.0001
	C416T	25	-	25/120	Non-synonymous	139 T to I	52.64	<0.0001
*CD14*	G457A	36	-	36/120	Non-synonymous	153 A to T	75.66	<0.0001
	A491C	19	-	19/120	Non-synonymous	164 Q to F	39.93	<0.0001
*CCL2*	A262G	27	-	27/120	Non-synonymous	88 R to G	56.74	<0.0001
	T479C	-	31	31/120	Non-synonymous	160 L to S	65.15	<0.0001
*TLR1*	A189T	-	26	26/120	Synonymous	63 A	54.64	<0.0001
	G285C	23	-	23/120	Non-synonymous	95 Q to H	48.34	<0.0001
	G308A	17	-	17/120	Non-synonymous	103 S to N	35.73	<0.0001
	G360C	38	-	38/120	Non-synonymous	102 E to D	79.86	<0.0001
	G394T	-	42	42/120	Non-synonymous	132 V to F	88.27	<0.0001
*TLR7*	C347T	36	-	36/120	Non-synonymous	116 A to V	75.66	<0.0001
*TLR8*	G709A	46	-	46/120	Non-synonymous	237 G to S	96.67	<0.0001
*TLR9*	A174G	-	28	28/120	Synonymous	58 G	58.84	<0.0001
*β defensin*	C30T	-	18	18/120	Synonymous	10 R	37.83	<0.0001
	T105A	-	49	49/120	Non-synonymous	35 D to E	102.98	<0.0001
	T197C	33	-	33/120	Non-synonymous	66 V to A	69.35	<0.0001
	C225T	-	24	24/120	Synonymous	75 T	50.44	<0.0001
*SP110*	A62C	27	-	27/120	Non-synonymous	21 N to T	56.74	<0.0001
	C184T	51	-	51/120	Non-synonymous	62 R to C	107.18	<0.0001
*SPP1*	C857T	-	36	36/120	Non-synonymous	286 T to I	75.66	<0.0001
*BP1*	A75G	16	-	16/120	Synonymous	25 T	33.63	<0.0001
	G257A	37	-	37/120	Non-synonymous	86 R to Q	77.76	<0.0001
	A456G	25	-	25/120	Synonymous	152 S	52.54	<0.0001
*A2M*	C36G	28	-	28/120	Synonymous	12 S	58.84	<0.0001
*ADORA3*	C387A	-	39	39/120	Synonymous	129 I	81.96	<0.0001
*CARD15*	A73G	-	24	24/120	Non-synonymous	25 S to G	50.44	<0.0001
*IRF3*	T133G	34	-	34/120	Non-synonymous	45 F to V	71.45	<0.0001
	C273G	23	-	23/120	Synonymous	91 S	48.34	<0.0001
*SCART1*	C42T	-	47	47/120	Synonymous	14 A	98.77	<0.0001
	A126G	19	-	19/120	Synonymous	42 S	39.93	<0.0001
	A147G	-	53	53/120	Synonymous	49 V	111.38	<0.0001
	T380C	-	31	31/120	Non-synonymous	127 V to A	65.15	<0.0001

*SLC11A 1* = solute carrier family 11 member 1; *CD1 4* = cluster of differentiation 14; *CCL2* = C-C motif ligand 2; *TLR1* = Toll-like receptor 1; *TLR 7* = Toll-like receptor 7; *TLR 8* = Toll-like receptor 8; *TLR 9* = Toll-like receptor 9; *β defensin* = beta defensin; *SP110* = SP110 nuclear body protein; *SPP1* = secreted phosphoprotein 1; *BP 1* = bactericidal/permeability-increasing protein; *A2M* = alpha-2-macroglobulin; *ADORA3* = adenosine A3 receptor; *CARD15* = caspase recruitment domain-containing protein 15; *IRF3* = interferon regulatory factor 3; *SCART1* = scavenger receptor family member expressed on T cells 1. A = alanine; C = cysteine; D = aspartic acid; E = glutamic acid; F = phenylalanine; G = glycine; H = histidine; I = isoleucine; L = leucine; M = methionine; P = proline; Q = glutamine; R = arginine; S = serine; T = threonine; and V = valine.

## Data Availability

Upon reasonable request, the corresponding author will provide the data backing up the study’s conclusions.

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
