# Peer review of "Individual Genomic Loci and mRNA Levels of Immune Biomarkers Associated with Pneumonia Susceptibility in Baladi Goats"

_vetsci, 2023, doi:10.3390/vetsci10030185_

Round 1

Reviewer 1 Report

This manuscript from Ateya et al details genetic analysis collected from a group of 120 Baladi goats in Egypt to screen for genetic variants associated with pneumonia.  Goats are a critical agricultural resource worldwide, and identifying genetic variants associated with diseases such as pneumonia may aid in selective breeding practices.  The group reports significant influence of SNPs in a number of immune-related genes in the development of pneumonia in these animals. 

The biggest problem to be addressed with this manuscript is that the RT-PCR analysis reported is not significant to the analysis of genetic variants.  The authors do not report the health of the animals at the time of sample collection, so the increased expression of inflammatory genes in pneumatic animals may be the result of clearing infection rather than genetic influence on gene expression.  As such, the authors would need a secondary method to confirm the influence of these genetic variants on pneumonia development, such as association of SNPs with long-term health or effects of drug treatment on animals with SNP variants.  

I have the following comments to also be addressed:

1) The quality of DNA reported in the methods section seems low for the reported yield of the kit- the authors should confirm DNA quality by including a PCR gel.

2) The SNPs in table 3 should also be plotted on a GWAS plot to show their significance amongst the wider data set.  The authors should also clarify whether there were no SNPs that were present in both healthy and pneumonic goats as the data suggests.

3) The authors should comment on the current capabilities of farms in Egypt to genetically screen their animals for selective breeding.

Author Response

Reviewer 1

Comment

This manuscript from Ateya et al details genetic analysis collected from a group of 120 Baladi goats in Egypt to screen for genetic variants associated with pneumonia.  Goats are a critical agricultural resource worldwide, and identifying genetic variants associated with diseases such as pneumonia may aid in selective breeding practices.  The group reports significant influence of SNPs in a number of immune-related genes in the development of pneumonia in these animals. 

The biggest problem to be addressed with this manuscript is that the RT-PCR analysis reported is not significant to the analysis of genetic variants. The authors do not report the health of the animals at the time of sample collection, so the increased expression of inflammatory genes in pneumatic animals may be the result of clearing infection rather than genetic influence on gene expression.  As such, the authors would need a secondary method to confirm the influence of these genetic variants on pneumonia development, such as association of SNPs with long-term health or effects of drug treatment on animals with SNP variants.  

Response

  • We thank reviewer for this comment. In the materials and methods section (part Animals and research subjects) we illustrated the health condition of the investigated animals by clinical examination and continuous observing for farm records as follows: A thorough clinical examination was performed on the goats under investigation in accordance with the previously described standard protocols, and the results were simultaneously recorded. Considering their health, the researched does were split into two equal-sized groups (60 each). Clinically healthy individuals made up the first group (i.e., normal appetite intake, body temperature, normal, physiological ranges of body temp, respiratory, pulse rates, bright eyes, and no lacrimal or nasal discharges) and was assigned as the healthy group. Pneumonia is present in the second group (mucopurulent nasal dis-charge, hyperthermia, abdominal respiration, weakness, off-appetite, crackles, exercise intolerance, gasp, and wheezes by auscultation).
  • It was established about gene expression analysis that, transcript abundance functions as a heritable endophenotype and is associated with chromosomal polymorphisms, in accordance with the genetic genomics theory. This method supported the idea that combining data on gene expression and chromosomal variants could aid in our understanding of the genetics underlying the onset of disease.
  • Our study was designed to overcome the limitations of previous work by investigating polymorphism in gene using SNP genetic marker and gene expression. Consequently, the investigated genes regulation mechanisms are well understood in the postpartum healthy and affected does.
  • In addition, previous studies used gene expression analysis as biomarker and for monitoring the health status of livestock. For instance the following;
  • Al-Sharif, M.; Ateya, A. New Insights on Coding Mutations and mRNA Levels of Candidate Genes Associated with Diarrhea Susceptibility in Baladi Goat. Agriculture 2023, 13, 143. https://doi.org/10.3390/ agriculture13010143
  • Saed HAR, Ibrahim HMM, El-Khodery SA, Youssef MA. Relationship between expression pattern of vitamin D receptor, 1 alpha-hydroxylase enzyme, and chemokine RANTES genes and selected serum parameters during transition period in Holstein dairy cows. Vet Rec Open. 2020;7(1):e000339.
  • Ateya A, El-Sayed A, Mohamed R. Gene expression and serum profile of antioxidant markers discriminate periparturient period time in dromedary camels. Mamm Res. 2021;66(4):603–613
  • Essa, B.; Al-Sharif, M.; Abdo, M.; Fericean, L.; Ateya, A. New Insights on Nucleotide Sequence Variants and mRNA Levels of Candidate Genes Assessing Resistance/Susceptibility to Mastitis in Holstein and Montbéliarde Dairy Cows. Vet. Sci. 2023, 10, 35. https:// doi.org/10.3390/vetsci10010035
  • Asmaa Darwish, Eman Ebissy, Ahmed Ateya & Ahmed El-Sayed (2021): Single nucleotide polymorphisms, gene expression and serum profile of immune and antioxidant markers associated with postpartum disorders susceptibility in Barki sheep, Animal Biotechnology, DOI: 10.1080/10495398.2021.1964984
  • Mona Al-Sharif, Basma H. Marghani & Ahmed Ateya (2022): DNA polymorphisms and expression profile of immune and antioxidant genes as biomarkers for reproductive disorders tolerance/susceptibility in Baladi goat, Animal Biotechnology, DOI: 10.1080/10495398.2022.2082975

I have the following comments to also be addressed:

Comment

  • The quality of DNA reported in the methods section seems low for the reported yield of the kit- the authors should confirm DNA quality by including a PCR gel.

Response

  • We are grateful to the reviewer for drawing it to our consideration. A gel electrophoresis is included in the supplementary material alongside Nano drop curve for ensuring good quality DNA.
  • We usually ensure the good quality DNA by making gel electrophoresis with Nano drop analysis.

Comment

  • The SNPs in table 3 should also be plotted on a GWAS plot to show their significance amongst the wider data set.  The authors should also clarify whether there were no SNPs that were present in both healthy and pneumonic goats as the data suggests.

Response

  • We thank reviewer for this comment. We examined the association between pneumonia incidence and nucleotide sequence variants of immune markers by PCR-DNA sequencing assessment. It is worth mentioning that, GWAS plot is indicative during application of genome wide analysis approach for looking forward to chromosomal regions containing genes controlling the target trait.
  • Although, genome wide association analysis provides information about the chromosomal regions containing target genes controlling the trait of interest; it lacks consistency in reporting single nucleotide polymorphisms (SNPs) implicated in susceptibility of disease. Consequently, our study’s main objective was to investigate the relationship between SNPs in candidate immune genes and the incidence of pneumonia resistance/susceptibility using PCR-DNA sequencing and real time PCR approaches. This ensures the effectiveness of investigated genes for selection of favorable or resistant animal.
  • Actually we search for presence or absence of SNPs in both healthy and pneumonic animals. To ensure that we use soft wares programs like CLUSTALW and MEGA; as indicated in our study. In addition previous studies used these programs for careful judgment presence or absence of SNPs as follows;
  • Al-Sharif, M.; Ateya, A. New Insights on Coding Mutations and mRNA Levels of Candidate Genes Associated with Diarrhea Susceptibility in Baladi Goat. Agriculture 2023, 13, 143. https://doi.org/10.3390/ agriculture13010143
  • Essa, B.; Al-Sharif, M.; Abdo, M.; Fericean, L.; Ateya, A. New Insights on Nucleotide Sequence Variants and mRNA Levels of Candidate Genes Assessing Resistance/Susceptibility to Mastitis in Holstein and Montbéliarde Dairy Cows. Vet. Sci. 2023, 10, 35. https:// doi.org/10.3390/vetsci10010035
  • Asmaa Darwish, Eman Ebissy, Ahmed Ateya & Ahmed El-Sayed (2021): Single nucleotide polymorphisms, gene expression and serum profile of immune and antioxidant markers associated with postpartum disorders susceptibility in Barki sheep, Animal Biotechnology, DOI: 10.1080/10495398.2021.1964984
  • Mona Al-Sharif, Basma H. Marghani & Ahmed Ateya (2022): DNA polymorphisms and expression profile of immune and antioxidant genes as biomarkers for reproductive disorders tolerance/susceptibility in Baladi goat, Animal Biotechnology, DOI: 10.1080/10495398.2022.2082975

Comment

  • The authors should comment on the current capabilities of farms in Egypt to genetically screen their animals for selective breeding.

Response

  • We are grateful to the reviewer for drawing it to our consideration. A signed consent form from the farm owner alongside an ethical approval is provided during submission of our manuscript according to criteria of ranked scientific journals.
  • The farm owner is informed about nature of the study and the project that the authors investigate.
  • The direct selection approach applied by animal breeders for disease resistance may be very difficult because of the low value of heritability and potentially low acting as selection criteria for resistant animal. However, utilization of a candidate gene approach explores if there are any genetic predisposing factors to the disease, by inspecting differences in the DNA known as single nucleotide polymorphisms (SNPs) in key genes associated with immune function .
  • Molecular techniques and genetic markers could help to overcome some of the confines of the traditional methods applied by animal breeder to improve animal health.
  • In fact, we tried in the last period to convince the breeders about the economic gain from application of marker assisted selection for selection of resistant animal and overcome the economic losses afforded by animal breeder. Actually, many breeders are convinced about genetic screening especially he can select the best animal at the early period of life even in the first months before entering the production period.
  • The breeders are convinced on the basis of they could overcome the huge economic losses from breeding of animals with expected unfavorable genotype; thus an early culling could be applied.

Reviewer 2 Report

Introduction. The first three paragraphs should be reduced in length, as they provide well-established knowledge.

The objectives of the work must be clearly defined and presented at the end of the section.

M & M

2.2. can be shortened.

Table 1. Please check carefully the primers, there is a slight error in one. The table should be transferred to supplementary material.

Table 2. Also, please transfer to supplementary material.

Results

Please insert a sub-section with correlations of clinical work with gene findings.

Discussion

This should be divided into sub-sections to allow better flow of ideas.

Author Response

Comments and Suggestions for Authors

Comment

Introduction. The first three paragraphs should be reduced in length, as they provide well-established knowledge.

Response

We are grateful to the reviewer for drawing it to our consideration. The first three paragraphs are reduced in in length.

Comment

The objectives of the work must be clearly defined and presented at the end of the section.

Response

Many cardinal thanks to the reviewer for this.  The objectives of the work are clearly defined and presented at the end of the introduction section.

M & M

Comment

2.2. can be shortened.

Response

Many cardinal thanks to the reviewer for this.  2.2. Part is shortened.

Comment

Table 1. Please check carefully the primers, there is a slight error in one. The table should be transferred to supplementary material.

Response

We thank the reviewer for this. Primers are carefully checked and the table is transferred to supplementary material.

Comment

Table 2. Also, please transfer to supplementary material.

Response

We thank the reviewer for this. Table 2 is transferred to supplementary material.

Comment

Results

Please insert a sub-section with correlations of clinical work with gene findings. 

Response

We thank the reviewer for this. A sub-section with correlations of clinical work with gene findings.

Comment

Discussion

This should be divided into sub-sections to allow better flow of ideas.

Response

We thank the reviewer for this. Discussion is divided into sub-sections to allow better flow of ideas.

Reviewer 3 Report

The article is original and very interesting. I suggest the following corrections

-1 editing corrections- you must use the Vet Sci Template, including the numbers of lines, to allow reviewer to make coments on some specific paragraphs

2. In Methodology section you must precise the etiology of pneumonia in your pneumonic goats (bacterial? which species? or viral?)

3 Not all the Figs should be supplementary materials. Those of Molecular biology, sustaining your results should be inserted into the manuscript

Author Response

Reviewer 3

Comments and Suggestions for Authors

Comment

The article is original and very interesting.

Response

Many cardinal thanks to the reviewer for this positive comment.

I suggest the following corrections

Comment

-1 editing corrections- you must use the Vet Sci Template, including the numbers of lines, to allow reviewer to make comments on some specific paragraphs.

Response

We are grateful to the reviewer for drawing it to our consideration. The manuscript is English edited by native English speaker provided by mdpi service.

Comment

  1. In Methodology section you must precise the etiology of pneumonia in your pneumonic goats (bacterial? which species? or viral?)

Response

  • We thank the reviewer for this. The main aim of our study is identifying the resistant genotype of pneumonia (I mean on the level of animal) otherwise the cause of pneumonia
  • It was established that pneumonia can be caused by a variety of causes, although the most frequent ones are bacterial or viral infections. Traditionally, Farm management practices and the creation of vaccines have always been the key priorities for pneumonia prevention. The use of pharmaceutical intervention in the treatment of pneumonia is significantly influenced by the source of the infection; for example, antibiotics are used to treat bacterially generated pneumonia.
  • Based on the aforementioned reasons, the main aim is identifying the favorable genotype for resistance (Marker assisted selection) to overcome the economic losses afforded by animal breeder.
  • Application of this study on the level of animal provides the economic gain from application of marker assisted selection for selection of resistant animal and overcome the economic losses afforded by animal breeder. Actually, genetic selection of the best animal at the early period of life even in the first months could be accomplished before entering the production period.
  • Whatever the cause, the breeders are convinced on the basis of they could overcome the huge economic losses from breeding of animals with expected unfavorable genotype; thus an early culling could be applied.
  • We anticipate that infectious pneumonia was the primary cause of the majority of the pneumonia cases in our study. Furthermore, our outcomes from Real Time PCR provide believable indication that animals with pneumonia experienced a strong inflammatory response.
  • The investigated genes are novel immune markers or scarcely reported; thus we tried to find an association between these markers and pneumonia resistance/susceptibility.
  • A further shortcoming will be applied on the levels of animal and the causative agent.

Comment

3 Not all the Figs should be supplementary materials. Those of Molecular biology, sustaining your results should be inserted into the manuscript

Response

We thank the reviewer for this. The supplementary data are easily provided in all mdpi journal by presenting a link for available screening and reading. Moreover, there are sixteen supplementary figures. If they are put in the body of the manuscript, it will be extra-long. Therefore, we are deeply indebted to the respected reviewer to leave this comment for authors’ consideration.

Round 2

Reviewer 1 Report

The authors have addressed most of my earlier comments, but I would suggest that they further clarify the following:

1) They should clarify whether DNA samples from pneumonic animals were taken during active disease or during convalescence.

2) The authors indicated that they tested for SNPs in healthy and pneumonic animals, but they should clarify whether any SNPs were found in both healthy and pneumonic animals, as table 1 suggests that each SNP was only found in healthy or pneumonic animals.

3) The authors should label the included agarose gel and include a positive and negative control to ensure specific amplification.

Author Response

Comment

They should clarify whether DNA samples from pneumonic animals were taken during active disease or during convalescence.

Response

We are grateful to the reviewer for drawing it to our consideration. The time of collection of blood samples for extraction of DNA and RNA is clarified. Actually blood samples were collected during the active period to obtain accurate results especially for real time PCR.

Comment

The authors indicated that they tested for SNPs in healthy and pneumonic animals, but they should clarify whether any SNPs were found in both healthy and pneumonic animals, as table 1 suggests that each SNP was only found in healthy or pneumonic animals.

Response

We are grateful to the reviewer for drawing it to our consideration. It is clarified according to suggestion in the results section (Section Polymorphisms of Immune markers)

Comment

The authors should label the included agarose gel and include a positive and negative control to ensure specific amplification.

Response

We thank reviewer for this. Labelling is done for the provided gel in supplementary files.

Reviewer 2 Report

Before acceptance, the authors can extend the discussion by adding some recent relevant references, to broaden the scope of the manuscript.

Author Response

Comment

Before acceptance, the authors can extend the discussion by adding some recent relevant references, to broaden the scope of the manuscript.

Response

We thank reviewer for this. The discussion is extended by adding some recent relevant references, to broaden the scope of the manuscript.

Reviewer 3 Report

I recommend the acceptance of the article in present form. The authors have made all the corrections suggested

Author Response

Comment

I recommend the acceptance of the article in present form. The authors have made all the corrections suggested

Response

Many cardinal thanks to the reviewer for this positive comment.